# Factors Influencing Gaseous Emissions in Constructed Wetlands: A Meta-Analysis and Systematic Review

**DOI:** 10.3390/ijerph20053876

**Published:** 2023-02-22

**Authors:** Sile Hu, Hui Zhu, Gary Bañuelos, Brian Shutes, Xinyi Wang, Shengnan Hou, Baixing Yan

**Affiliations:** 1Key Laboratory of Wetland Ecology and Environment, Northeast Institute of Geography and Agroecology, Chinese Academy of Sciences, Changchun 130102, China; 2College of Resources and Environment, University of Chinese Academy of Sciences, Beijing 100049, China; 3USDA, Agricultural Research Service, San Joaquin Valley Agricultural Science Center, 9611 South Riverbend Avenue, Parlier, CA 93648-9757, USA; 4Department of Natural Sciences, Middlesex University, Hendon, London NW4 4BT, UK

**Keywords:** constructed wetlands, GHG emission, NH_3_ volatilization, VOCs, H_2_S, meta-analysis

## Abstract

Constructed wetlands (CWs) are an eco-technology for wastewater treatment and are applied worldwide. Due to the regular influx of pollutants, CWs can release considerable quantities of greenhouse gases (GHGs), ammonia (NH_3_), and other atmospheric pollutants, such as volatile organic compounds (VOCs) and hydrogen sulfide (H_2_S), etc., which will aggravate global warming, degrade air quality and even threaten human health. However, there is a lack of systematic understanding of factors affecting the emission of these gases in CWs. In this study, we applied meta-analysis to quantitatively review the main influencing factors of GHG emission from CWs; meanwhile, the emissions of NH_3_, VOCs, and H_2_S were qualitatively assessed. Meta-analysis indicates that horizontal subsurface flow (HSSF) CWs emit less CH_4_ and N_2_O than free water surface flow (FWS) CWs. The addition of biochar can mitigate N_2_O emission compared to gravel-based CWs but has the risk of increasing CH_4_ emission. Polyculture CWs stimulate CH_4_ emission but pose no influence on N_2_O emission compared to monoculture CWs. The influent wastewater characteristics (e.g., C/N ratio, salinity) and environmental conditions (e.g., temperature) can also impact GHG emission. The NH_3_ volatilization from CWs is positively related to the influent nitrogen concentration and pH value. High plant species richness tends to reduce NH_3_ volatilization and plant composition showed greater effects than species richness. Though VOCs and H_2_S emissions from CWs do not always occur, it should be a concern when using CWs to treat wastewater containing hydrocarbon and acid. This study provides solid references for simultaneously achieving pollutant removal and reducing gaseous emission from CWs, which avoids the transformation of water pollution into air contamination.

## 1. Introduction

Constructed wetlands (CWs) are wastewater treatment systems that mimic the purification function of natural wetlands through physical, chemical, and biological processes. As a typical nature-based solution, CWs have been widely applied worldwide for treating various wastewater, including municipal wastewater, domestic sewage, agricultural runoff, industrial wastewater, etc. [1,2]. It is foreseeable that there will be a rapid increase in newly established CWs in the future, especially in developing countries and rural areas, due to their high efficiency of pollutant removal, stable effluent water quality, low investment, and simple maintenance.

As a result of plant respiration, microbial assimilation and decomposition, diffusion, and ebullition, some elements in the wastewater are released into the atmosphere in gaseous form during pollutant removal. The main gases emitted from the wastewater treatment systems include greenhouse gases (GHGs) such as carbon dioxide (CO_2_), methane (CH_4_), and nitrous oxide (N_2_O) and odorant substances including ammonia (NH_3_), volatile organic compounds (VOCs), and hydrogen sulfide (H_2_S) [3]. Therein, CO_2_ is one of the main GHGs that cause climate change. Methane and N_2_O, as more potent GHGs, have 28 and 298 times more global warming potential (GWP) than CO_2_ [4,5]. As the primary alkaline molecule, NH_3_ plays a crucial role in fine particle pollution, acidification, and nitrogen deposition, which have been causing global climate change and even harming human health [6]. VOCs are a diverse group of substances that can react with nitrite and hydroxyl radicals, increasing ozone (O_3_) concentrations in the lower troposphere and forming organic aerosols. In addition to the gases mentioned above, people are also concerned with H_2_S due to its odorous and hazardous properties and its potential contribution to the acidification of ecosystems [7]. For these reasons, there has been a growing concern about gas emissions from wastewater treatment systems.

Compared to wastewater treatment plants, CWs are reported to emit relatively lower CO_2_, CH_4_, and N_2_O [8,9]. However, a previous study showed that CWs emitted 2–10 times more GHGs than natural wetlands [10]. Additionally, the NH_3_ volatilization reportedly contributed to 44% of the total ammoniacal nitrogen (TAN) removal in free water surface flow (FWS) CWs [11]. In a marsh–pond–marsh system, NH_3_ volatilization can reach up to 79% of the nitrogen removal when the nitrogen loads are greater than 15 kg/ha/d [12]. In addition, VOCs and H_2_S emissions occur in CWs treating wastewater, such as petroleum-contaminated groundwater and acidic drainage, respectively. Therefore, it is necessary to be concerned about the various gas emissions from CWs, thereby preventing the transformation of water pollution into air pollution.

The biological, chemical, and physical processes of pollutant mitigation and transformation can be affected by various factors such as CW types, substrate types, wetland plants, wastewater characteristics, environmental conditions, etc., resulting in different levels of gas emissions. Therefore, it is essential to comprehend how these factors impact these gas emissions. Maucieri et al. [13] reviewed the research progress of GHG (CH_4_, N_2_O, and CO_2_) emissions from CWs. They applied a non-weighted meta-analysis to analyze a few influencing factors including CW typologies, wastewater composition, and plant species using data obtained from 29 papers published from 1997 to 2016. Their study demonstrated that the following: the CH_4_ emissions from subsurface flow (SSF) CWs were lower than FWS CWs; the N_2_O emissions from vertical subsurface flow (VSSF) CWs were lower than FWS CWs; the influent COD/N ratio of 5 was optimal to obtain the lowest N_2_O emission from CWs; plant presence significantly increases the CO_2_ emission compared to all unvegetated CW types and increases N_2_O and CH_4_ emissions in VSSF CWs; and no specific result was found about the impact of plant species richness on CH_4_ emission from CWs. However, with the globally increasing concern for carbon reduction, plenty of new relevant publications have arisen in recent years. Will the multiplied new literature still support the original cognition or provide new scientific conclusions? This remains unclear. Furthermore, except for the GHG emissions, there is no systematic review of the other gas (i.e., NH_3_, VOCs, and H_2_S) emissions, which limits the complete understanding of the comprehensive performance and possible impact of CWs.

To fill the above knowledge gaps, this study focused on the main influencing factors of multiple gaseous emission from CWs. Our study applied meta-analysis to quantitatively analyze the main influencing factors of GHG emission using data extracted from 63 publications from 1997 to 2021. Meanwhile, we qualitatively analyzed the influencing factors of NH_3_, VOC, and H_2_S emissions from CWs. Based on the conclusions obtained from the current research progress, recommendations for future study are also provided. The new scientific evidence and understanding obtained in this study will provide a reference for optimizing the design and operation of CWs, considering their ecological and environmental benefits.

## 2. Materials and Methods

The literature survey of peer-reviewed publications about CWs and gaseous (i.e., GHGs, NH_3_, VOCs, H_2_S) emission was carried out using the Web of Science. We collected the publications focused on gaseous emission from CWs and the factors that possibly impact the emission of gases. Due to the limited number of publications regarding a specific factor and/or a specific gas, not all publications conform to quantitative analysis. Therefore, the literature was analyzed considering two approaches: (1) a quantitative review with meta-analysis updates the information about how CW types, substrate types, and plant species and richness impact GHG emission; (2) a qualitative review provides a synthetic conclusion about factors affecting the NH3, VOC, and H_2_S emissions, as well as some specific factors affecting the GHG emission. The data processing of meta-analysis is described below in Section 2.1 and Section 2.2.

### 2.1. Literature Search and Data Collection Criteria

We initially searched the publications published before 23 November 2021 using Web of Science. The search term was TS = (“constructed wetland*” OR “artificial wetland*” OR “treatment wetland*”) AND TS = (“CO_2_” OR “carbon dioxide” OR “N_2_O” OR “nitrous oxide” OR “methane” OR “CH_4_” OR “GHG*” OR “greenhouse gas*.”)

The following criteria were used to select relevant publications: (1) select the publications focusing on the factors likely to affect GHG emission, e.g., the structure of CWs, wastewater characteristics and environmental conditions, etc.; (2) the published data must include at least one treatment group compared to a control group involved in the factors; (3) if the publications included different CW types, substrate types, plant species, and other factors, the observations were classified as independent results; (4) If the information involved unvegetated and vegetated systems, we chose the monoculture system compared to the unvegetated systems, and comparisons between polyculture systems containing various plant species and unvegetated systems were excluded; (5) the aquatic plants mainly refer to emergent and floating macrophytes, and trees and bushes (e.g., mangrove) were excluded; (6) to ensure the sample size of substrate types, studies using various types and sizes of gravel were all considered as a gravel-based system. The literature based on the search term described above showed 979 results. Finally, 63 peer-reviewed publications were used in this meta-analysis, and these publications came from 12 countries according to the first affiliation of the corresponding author, including China, Canada, Japan, etc. (Figure 1). The flowchart of the selection process is displayed in Appendix A. The other publications were selectively used for the qualitative analysis of the GHGs from CWs.

### 2.2. Data Extraction and Calculation

Data were extracted directly from the text, tables, and figures from publications using GetData Graph Digitizer 2.24.

Meta-analysis was widely adapted to summarize experimental behavioral, medical, and social sciences evidence. The conventional effect size considers the logarithm ratio of the treatment value to the control value, but this effect size was not applicable in this study because there were negative GHG fluxes in our chosen studies. Therefore, we applied the index that was used in previous publications [13]:(1)$$T_{GHG}=\pm \frac{\left|GHG_{t}-GHG_{c}\right|}{\left|GHG_{t}\right|+\left|GHG_{c}\right|}$$
where *GHG_t_* and *GHG_c_* indicate individual, paired GHG values of the control and treatment group. The variable *T_GHG_* is assigned a sign based on the direction of difference (if any) in GHGs flux due to the treatment group activity: a negative *T* indicates a decrease in GHGs with the treated activity; conversely, a positive T indicates an increase in GHGs. The index ranges from −1 to +1, and an index of zero indicates no difference between the control and treatment groups.

We conducted an unweighted meta-analysis using GraphPad to calculate the 95% bias-corrected confidence intervals (95% CIs). It was considered significant (at *p* < 0.05) when 95% CIs did not overlap with zero [14]. The t-tests and one-way ANOVA were applied to test for the differences in variables.

## 3. Results and Discussion

### 3.1. Factors Affecting GHG Emission from CWs

#### 3.1.1. CW Types

Constructed wetlands can be generally separated into three typical types, i.e., FWS CWs, horizontal subsurface flow (HSSF) CWs, and VSSF CWs, which were distinguished by their structure and water flow direction. The water in both FWS and HSSF CWs flows horizontally, but they differ in water level. A typical FWS CW with emergent macrophytes is a shallow sealed basin or sequence of basins, including 20–30 cm of rooting soil (or the other medium), with a water depth of 20–40 cm. The HSSF CWs commonly consist of gravel or rock beds and are vegetated with wetland plants. The influent wastewater flows through the substrate medium under the surface of the bed in a more or less horizontal path [15]. The VSSF CWs comprise a substrate of gravel or coarse sand planted with macrophytes. The wastewater in VSSF CWs distributes uniformly at the surface of the wetland and then flows vertically to the bottom [16]. Different CWs provide aerobic and/or anaerobic zones (which mainly impact the formation and oxidation of CH_4_, nitrification, and denitrification) due to differences in water levels and flow patterns. The FWS CWs provide both aerobic and anaerobic zones. The anoxic and anaerobic processes are prevalent in HSSF CWs and relatively fewer aerobic zones appeared in VSSF CWs [15]. The various CWs provide different conditions to impact the processes of pollutants’ migration and transportation in CWs, consequently affecting GHG emission. Based on the meta-analysis, the HSSF and VSSF CWs emit less CH_4_ than FWS CWs (Figure 2a), but there was no significant difference between HSSF and VSSF CWs (*t*-test, *p* > 0.05). The HSSF CWs decrease the N_2_O emission as compared to VSSF and FWS CWs, and a statistically significant difference (*t*-test, *p* < 0.05) was observed between HSSF and VSSF CWs (Figure 2b).

Microbial community structure and activity are different in various types of CWs. The biochemical processes in CWs are driven by different oxidation-reduction potentials (ORP). The ORP can be divided into two levels, favored oxidative (aerobic) and reductive (anaerobic) conditions, ranging from +250 to +700 mV and +250 to −400 mV, respectively [16]. Though the total activity of the microbial community is in the same range in different CW types (different CWs provide different aerobic or anaerobic conditions), structural differences exist in microbial communities [17]. Therefore, the difference in the microbial community under varying conditions (i.e., different ORP values) may cause various CH_4_ and N_2_O emissions.

Methane is produced by methanogens during the mineralization of organic matter, which occurs in anaerobic conditions. Gui et al. [18] found that areas with ORP < −100 mV (i.e., anaerobic) were mainly located at the sand surface in FWS CWs but were not found in VSSF CWs. This observation can explain the lower CH_4_ emission in VSSF CWs compared to FWS CWs. In HSSF CWs, the aerobic area was mainly concentrated near the surface, when CH_4_ diffused from the bottom to the surface, and it is degraded simultaneously [18]. Therefore, HSSF and VSSF CWs can reduce CH_4_ emission compared to FWS CWs.

The nitrification and denitrification processes are considered the main source of N_2_O in CWs. These processes occur in all types of CWs and the extent of the process is determined by the available O_2_ [19]. Compared to FWS CWs, subsurface (SSF) CWs provide more anoxic conditions for denitrifying bacteria to grow. Especially in HSSF CWs, prevailing anoxic and anaerobic conditions provide suitable conditions for denitrification [20], while VSSF CWs are more aerobic and provide little denitrification compared to HSSF CWs [21]. In low dissolved oxygen (DO) conditions, N_2_O will be replaced by nitrogen gas (N_2_) as the end product of denitrification and can be considered as an N_2_O sink in wastewater treatment [22]. These above findings possibly explain the lower N_2_O emission in HSSF CWs compared to FWS and VSSF CWs, as illustrated in Figure 2b.

Both HSSF and VSSF CWs could effectively remove pollutants such as BOD (biochemical oxygen demand) and chemical oxygen demand (COD) [23]. The VSSF CWs are preferable to remove ammonia nitrogen (NH_4_^+^) than FWS and HSSF CWs, but exhibited poor nitrate (NO_3_^-^) removal ability [24]. A previous study showed that no significant difference was found between the TN removal of FWS and SSF CWs [15]. The SSF CWs are commonly used as the secondary treatment stage while FWS CWs are used as the tertiary stage. Overall, SSF CWs tend to perform better pollutant removal efficiency (e.g., organic pollutants) than FWS CWs. Although both HSSF and VSSF CWs can reduce CH_4_ emission, there is no statistical difference between them. In addition, HSSF CWs significantly reduce N_2_O emission compared to VSSF CWs as indicated in our meta-analysis. Therefore, HSSF CWs are recommended as the optimal wetland type when considering pollutant removal capacity and GHG emission.

#### 3.1.2. Substrate Types

The substrate plays an essential role in CWs. It acts as a block and filter to remove pollutants, supports reactive conditions for pollutant transformation, and provides a surface for biofilm attachment [25]. Hence, different filling materials in CWs exhibited distinct performances due to their specific physical and chemical properties. Substrate materials provide habitats for microbial communities. Microbes attached to the substrates can harbor functional genes related to the conversion of N_2_O and N_2_ (e.g., *nosZ*, *nirK*, and *nirS*) and the production and consumption of CH_4_ (e.g., *mcrA* and *pmoA*) [25,26]. Thus, selecting the proper substrates can potentially alleviate the GWP via reducing N_2_O and CH_4_. To date, various substrates have attracted attention for their impact on greenhouse gas emissions, such as biochar, manganese ore, iron ochre, zeolites, activated alumina, etc. Therein, biochar has received considerably more attention than other substrate materials, and meta-analysis was conducted on biochar considering the number of publications and sample size. The meta-analysis shows that the biochar-amended CWs can alleviate the release of N_2_O compared to gravel-based CWs (Figure 3a). However, biochar tends to increase the emission of CH_4_ (Figure 3b).

Biochar, a porous carbon-rich material produced from biomass under limited temperature and O_2_ conditions, is considered a promising amendment in CWs and displays both a potent water purification ability and satisfactory GHG reduction effectiveness. Several studies demonstrate that the biochar-amended CWs reduce N_2_O emission compared to the gravel-based CWs [27,28,29]. The biochar-amended system also showed higher nitrogen removal efficiency than the gravel-based system due to the better adsorption capacity of biochar [29]. The adsorption of NO_3_^-^ or/and NH_4_^+^ by biochar can decrease the substrate availability of nitrification and denitrification then suppress the production of N_2_O [30]. The large surface area and high porosity of biochar provide more attachment sites for abundant microbes and the carbon source released from biochar ultimately facilitates denitrification [28]. Biochar addition also regulates the microbial community structure and the abundance of functional genes. For example, biochar-amended CWs reduce the N_2_O emission by increasing the abundance of various bacteria (e.g., phyla of *Proteobacteria*, *Firmicutes*, *Bacteroidetes*, *Actinobacteria*, *Chloroflexi*) mainly participating in the denitrification process (mainly acts on the conversion of N_2_O to N_2_) [31,32]. Furthermore, biochar addition can increase the abundance of some bacteria harboring the *nosZ* gene that promote the biotransformation of N_2_O to N_2_ and the ratio of *nos*Z/ (*nirS* + *nirK*), which can partially explain the role of biochar in reducing N_2_O emission [25,33].

As illustrated in Figure 3b, biochar-amended CWs increased CH_4_ emission compared to gravel-based CWs. This result is probably because biochar had a high redox-active property or charging and discharging capacities, which can promote the direct interspecies electron transfer between methanogens and anaerobic bacteria (e.g., *Geobacteraceae*) and provides labile organic carbon for CH_4_ production [34]. When considering multiple GHGs, biochar-amended CWs commonly showed lower GWP than gravel-based CWs. For example, an 18–24% lower GWP in biochar-amended CWs than that of gravel-based CWs was reported by Guo et al. [29]. The capacity of biochar addition in mitigating GWP was also impacted by different types of CWs. In general, biochar-amended SSF CWs are more capable of mitigating GWP compared to FWS CWs [33].

Other substrate materials have also received attention in regards to GHG emission from CWs (Appendix A). A few studies showed that the addition of manganese ore (Mn ore) and iron ore can reduce CH_4_ emission in CWs [35,36,37]. A high abundance of *pmoA* and low *mcrA* was observed in Mn-ore-amended CWs, indicating that Mn ore not only inhibits the CH_4_ production but also intensifies the CH_4_ oxidation. The manganese oxides can hinder the production of CH_4_ by competing for organic substrates and promoting the anaerobic oxidation of methane by providing electron acceptors [38]. Considering the comprehensive impacts on multiple GHGs in CWs, Mn-ore-amended CWs exhibited lower GWP than CWs filled with gravel and iron ore, respectively [39]. Mn ore was favorable for mitigating GWP compared to CWs amended with walnut shell and activated alumina [38]. However, there is limited research on the substrate materials that concurrently reduce both CH_4_ and N_2_O emissions from CWs. Hence, the use of substrate materials on reducing GHGs should be explored in the future.

Overall, biochar-amended CWs are favorable for enhancing the removal performance and mitigating GWP compared to gravel-based CWs. Biochar is easily available and can be produced from multifarious biomass such as agricultural residues, woods, livestock manures, and solid organic municipal wastes, which can simultaneously solve the problem of agricultural waste. Furthermore, biochar has already been applied in wastewater treatment and soil remediation [40,41]. Therefore, biochar is suggested to be a promising amendment for future CW design.

#### 3.1.3. Plant Presence, Species, and Richness

Plants are an important component of CWs, and the presence of plants distinguishes CWs from lagoons or unvegetated filters. Plants play multiple roles in CWs, e.g., absorbing various pollutants, increasing substrate porosity, providing O_2_, and supporting a favorable environment for microbes [42]. Meanwhile, plant presence can directly or indirectly affect the transformation of pollutants and consequently impact the emission of gases in CWs. Except for O_2_ and CO_2_, gases emitted from wetlands have been commonly assumed to be generated from the sediment or soil surface through the water column, escape to the atmosphere by bubbling or diffusing called ebullition, or transport through the vascular system of emergent plants [43,44]. The plant species and combination can affect the contamination balances, and then impact removal efficiency and gas emission [45].

Influence of plants on CH_4_ emission

The available data from the literature show that the presence of plants reduces the CH_4_ emission from HSSF CWs but increases CH_4_ emission from FWS and VSSF CWs (Figure 4a–c). The structure of CWs can impact the activity of methanogens and methanotrophs due to the different redox conditions mentioned above. The presence of plants may further affect these microbes in different CWs, resulting to a varying extent in CH_4_ release. In vegetated HSSF CWs, the plant can transport O_2_ to the rhizosphere and promote the activity of methanotrophs, which is beneficial for the oxidization of CH_4_ [46]. In contrast, the unvegetated HSSF CWs are more conducive to methanogens’ inhabitation than vegetated HSSF CWs. As compared to HSSF CWs, FWS and VSSF CWs are more aerobic. There is speculation that the plant tissue can emit CH_4_ under aerobic conditions [47]. Plants can act as gas conduits for CH_4_ passing through and facilitating the gas transport from the sediment to the atmosphere [48]. These findings might explain the higher CH_4_ emission in vegetated FWS and VSSF CWs compared to unvegetated FWS and VSSF CWs.

Plant species can impact CH_4_ emission by changing the microbial communities and biochemical pathways for methanogenesis [49,50,51]. Different plant species can affect the microenvironment in CWs, as their plant roots release different levels of O_2_ and exudates [52]. For example, *Phragmites australis* is reported to restrain the activity of methane-producing microorganisms (e.g., methanogens) and promote the activity of methanotrophs in the root zone at a greater depth [53]. High proportions of gas space were found in rhizomes of *P. australis*, which supply the O_2_ demand in the rhizospheric environment, probably causing the attachment of numerous bacteria to the root surface (34–37% to be potential methane-oxidizing bacteria) [46]. Other plant species, e.g., *Zizania latifolia*, contributed to a more reductive status (i.e., the anaerobic condition) and higher biomass of methanogens (high level of CH_4_) [53]. The inhibition of methanogens and promotion of the methanotroph with different plants in CWs may be attributed to their O_2_ transformation ability. These results suggest that different plant species develop specific microbial communities in the rhizosphere, and the CH_4_ emission from CWs tends to be plant-species-specific.

Plant species richness also impacts CH_4_ emission from CWs. Meta-analysis shows that, irrespective of the number of plant species, polyculture CWs stimulate the CH_4_ emission compared to monoculture CWs (Figure 5) and there is no significant difference among the different combinations of various species (one-way ANOVA, *p* > 0.05). Higher amounts of methanogens and lower amounts of methanotrophs were observed in polyculture systems [53,54,55], which can partially explain the higher CH_4_ in polyculture systems since methanogens and methanotrophs and their activities play a dominant role in CH_4_ production and consumption in CWs. Notably, although studies support the emission of CH_4_ being positively related to methanogen numbers in polyculture CWs, Zhang et al. [47] found a negative correlation between CH_4_ emission and methanogen numbers and activity in polyculture CW microcosms, implying that plant tissues may also emit a certain amount of CH_4_ in aerobic conditions.

Influence of plants on CO_2_ emission

As presented in Figure 6a–c, the presence of plants promotes CO_2_ emission irrespective of CW types. In CWs, the sequestration and emission of CO_2_ are related to plant photosynthesis, respiration, soil respiration, and mineralization. In wastewater treatment, it is an inevitable procedure to convert organic matter into CO_2_. Vegetated CWs exhibited higher CO_2_ emission than the unvegetated CWs [56]. The cumulative CO_2_ efflux in vegetated CWs was higher than that of unvegetated units, indicating that the plant presence was significant for ecosystem respiration [57]. Wu et al. [58] demonstrated that CO_2_ flux in FWS CWs was also affected by plant species, and the highest net CO_2_ flux appeared in the *Zizania caduciflora* system, followed by the *P. australis* and *Cyperus rotundus* systems. In general, planted CWs tend to emit higher amounts of CO_2_ compared to unplanted systems and the extent of this promotive effect depends on plant species and their respective individual differences in growth characteristics and ecotypes.

Influence of plants on N_2_O emission

Nitrogen removal in CWs is accomplished by plant uptake, microbial activities (e.g., assimilation, nitrification, denitrification, and ammonification), decomposition, sedimentation, and volatilization [52]. The available nitrogen, carbon, and DO in water are major parameters involved in nitrogen transformation that can impact N_2_O emission from CWs. Wetlands plants are the primary drivers to change these parameters and affect N_2_O emission [59].

The result of meta-analysis suggests that the presence of plants decreases N_2_O emission in HSSF CWs but increases N_2_O emission in FWS and VSSF CWs (Figure 7a–c). The HSSF CWs provide more anaerobic conditions and favor the denitrification process compared to FWS and VSSF CWs. The presence of plants in CWs can provide carbon required for metabolic activities such as denitrification [60]. When the influent C/N ratio is insufficient to support denitrification, the available carbon from plants may be a more important factor than the O_2_ level to control the denitrification process and reduce N_2_O production. This process may partially explain the lower N_2_O production in vegetated HSSF CWs than in unvegetated HSSF CWs. However, when the influent C/N ratio is adequate, the O_2_ level may be the dominant factor controlling denitrification. The aerobic conditions (high level of O_2_) inhibit the denitrification and anaerobic conditions promote the conversion of N_2_O to N_2_. In unvegetated FWS and VSSF CWs, anaerobic conditions are prevailing. N_2_O can become a substitute for the electron acceptor and be consumed by microbes (e.g., denitrifying bacteria). Conversely, the vegetated FWS and VSSF CWs provide more aerobic conditions, and the consumption of N_2_O is low due to the high utilization rate of NO_3_^-^ produced by the nitrification of ammonia [61].

Some studies suggested that high plant species richness tends to reduce N_2_O emission in CW microcosms [62,63], while others found that high plant diversity tends to enhance N_2_O emission [59,64]. In this study, meta-analysis indicates that polyculture CWs basically do not influence N_2_O emission compared to monoculture CWs (Figure 8). Although there is no consensus from previous studies regarding the effectiveness of plant species richness on N_2_O production, most of the available studies concur that specific plant species in the CWs, regardless of monoculture or polyculture, significantly impact N_2_O emission, resulting in a lower or higher flux of N_2_O [59,65,66].

Irrespective of monoculture or polyculture CWs, the different levels of N_2_O emission are possible because the plant species can impact the activity of ammonia-oxidizing bacteria (AOB), denitrifying bacteria, and the nitrification/denitrification process. The root structure of specific plant species can enhance or inhibit the AOB and influence N_2_O production [67,68]. Plant litter and detritus also exhibit species-specific differences [69]. For example, *Cyperus alternifolius* can exude more organic carbon from roots and utilize organic carbon as a source for denitrifying bacteria, which promotes the production of N_2_O [65]. The developed aerenchyma of plants also (e.g., *P. arundinacea*) results in the emission of high amounts of N_2_O due to incomplete denitrification [59]. However, different plant species tend to develop different types of denitrifying bacteria. For example, *Rhizobium*, *Rhodopseudomonas*, *Bradyrhizobium*, *Rhodanobacter*, *Rubrivivax*, and *Pseudomonas* were reported as dominant denitrifying bacteria in CWs planted with *P. australis*, *Typha angustifolia*, and *C. alternifolius* [70]. In contrast, *Bradyrhizobium*, *Bosea,* and *Rhodopseudomonas* were dominant denitrifying bacteria in CWs planted with *Iris pseudacorus*, *Canna glauca*, *Scirpus Validus*, and *C. alternifolius* [71]. Different plants may impact the denitrifying bacteria that participate in the reduction of NO_3_^-^ to N_2_O or the conversion of N_2_O to N_2_ processes, leading to different amounts of N_2_O. In general, the N_2_O emission flux is plant-species-specific. We suggest that selecting the specific species is more important than increasing the species richness.

#### 3.1.4. Influent Water and Environmental Conditions

Increased nutrients and organic matter input to CWs increase the productivity of a wetland ecosystem and increase the production of GHGs [72]. In addition to this direct influence of carbon and nitrogen input on GHG emission from CWs, an increasing number of studies have focused on the characteristics of influent wastewater (e.g., influent C/N ratio and salinity) and environmental conditions (e.g., temperature).

The influent C/N ratio remarkably impacts the pollutant removal efficiency and GHG emission from CWs. Most studies support the conclusion that a C/N ratio of 5 is optimal for removing COD and total nitrogen (TN) [73,74] and exhibited lower GHG emission [75,76,77]. Chen et al. [78] reported that the removal efficiency of NO_3_^-^ increased with the increasing C/N ratio, but the TN removal percentage did not follow this trend. The high COD concentrations may inhibit the metabolism of microbes, resulting in decreased TN removal. For GHGs, most studies have focused on the influence of C/N on N_2_O emission, and only a few studies were concerned about the CO_2_ and CH_4_ emissions. When converted to CO_2_-equivalent, the contribution of N_2_O to total GHG emission was significant when C/N ratio ranged from approximately 1:2 to 1:4. However, N_2_O had a small contribution when C/N ratio was between about 3:1 and 8:1 [79]. The smaller N_2_O contribution under a C/N ratio of 10 was also reported by Chen et al. [78]. The relatively lower C/N ratio caused incomplete denitrification by denitrifiers (lacking the *nosZ* gene encoding N_2_O-reductase- ), resulting in increased N_2_O as an end product of denitrification [80]. In contrast, denitrification processes enhanced by the communities of denitrifiers (with the *nosZ* gene) under a higher C/N ratio (i.e., 5 and 10) can inhibit N_2_O release [75,78]. As for CO_2_ emission, it appears to be positively related to the influent carbon [77]. The high CO_2_ emission was attributed to the efficient degradation of influent carbon facilitated by an adequate O_2_ supply [81]. Compared to N_2_O, the contribution of CH_4_ to the total GHGs appears to be insignificant under different C/N ratios in SSF CWs [77,78]. These observations can be explained by the inhibited activity of methanogens under aerobic conditions or some of the CH_4_ generated in anaerobic conditions was oxidized to CO_2_ by methanotrophs in SSF CWs.

Some CWs are also applied to treat saline wastewater originating from sources including aquaculture, agriculture, and industrial sectors [82]. High salinity in wastewater impacts the function of microorganisms and plants, causing different pollutant removal capacities and GHG emissions in CWs. Therefore, it is important to be concerned about the impact of salt stress on CWs. High salinity tended to reduce pollutant removal efficiency and inhibit CO_2_ and CH_4_ emission from CWs via inhibiting plant growth and microbial communities (e.g., methanogens) [83,84]. A similar observation was reported by Poffenbarger et al. [85] that CH_4_ fluxes decreased from tidal marshes with increasing salinity. The response of N_2_O emission in CWs to salt stress remains limited, and there is no consensus from natural wetlands for reference, as both increases and no changes of N_2_O emission were reported in response to salinity [84,86,87]. Further research on N_2_O emission need to be investigated under different saline conditions, as well as the driven factors.

GHG emission from CWs is greatly affected by natural conditions, especially temperature. The production of GHGs by photosynthesis and heterotrophic microbial activities varies with temperature. The CH_4_ emission is higher in spring and summer than in fall and winter [88,89]. The increasing temperature stimulates plant synthesis and microbial activities (e.g., methanogens, etc.), resulting in more CH_4_ emissions from CWs [49,88]. In contrast, low temperatures (e.g., below 10 °C) tend to inhibit the development of plants and microbial activity resulting in CH_4_ decline [90]. With the seasonal temperature changes, CH_4_ emission was impacted by the transport potential of plants (e.g., *P. australis* and *T. latifolia*) [91]. Higher N_2_O emission also appeared in summer [67]. The high temperature may increase the abundance of nitrifying and denitrifying bacteria, leading to more N_2_O emission. In contrast, the plant withering and inactive microbial activity caused low N_2_O emission in winter [92]. In general, changes in seasonal temperature impact the development of plants and microbial communities and lead to different levels of CH_4_ and N_2_O emissions.

In summary, GHGs from CWs are affected by various factors including the structure of CWs (i.e., CW types, substrates, and plants), influent water parameters (e.g., C/N ratio, salinity), and environmental temperature. In view of the demand for carbon emission reduction targets, we suggest that the factors mentioned above should be optimized in the design and operation of CWs in the future. Additionally, it is promising to alleviate GHG emission by altering the microbial processes within CWs. Recent research implied that enhancing the anammox (anaerobic ammonium oxidation) and n-damo (nitrite-dependent anaerobic methane oxidation) processes may mitigate both CH_4_ and N_2_O emissions from CWs [93]. Anammox is an anerobic process that is different from denitrification—it is without N_2_O production and requires no external carbon source. The n-damo process serves as an essential CH_4_ and nitrogen sink and n-damo bacteria could utilize NO_3_^-^ as an electron acceptor to produce N_2_ and CO_2_, inhibiting the production of N_2_O and CH_4_. To date, lab-scale studies found that anammox and n-damo processes can reduce N_2_O and CH_4_ emission to some extent [94,95]. In addition, n-damo archaea and anammox bacteria abundance can increase over time in CWs sediment, suggesting that the cooperation between two microbial processes completes the denitrification process from NO_3_^-^ to N_2_ [96,97]. However, it is essential to investigate the optimum condition to culture two microorganisms and ensure their colonization within CWs.

### 3.2. Factors Affecting NH_3_ Volatilization from CWs

NH_3_ volatilization is an unexpected but main pathway to remove nitrogen in CWs especially for treating swine wastewater and agricultural wastewater, and it has received much attention in recent decades [11,98]. NH_3_ volatilization in different CWs has been reported to contribute a minimum of 1.7% and a maximum of 44% of nitrogen removal [11,99,100]. In a marsh–pond–marsh system, the NH_3_ volatilization can even reach up to 79% of the nitrogen removal when the nitrogen load is greater than 15 kg/ha/d [12]. In general, NH_3_ volatilization occupies a large proportion of nitrogen removal in specific conditions. Therefore, it is essential to understand the mechanism and influencing factors of NH_3_ volatilization in CWs.

As a soluble gas, NH_3_ is directly transferred across the water–air interface, the surface partial pressure of which is a function of Henry’s Law (i.e., equilibrium between NH_3_ and NH_4_^+^), rather than ebullition and vegetative transport [11]. In CWs, NH_3_ volatilization is mainly controlled by wastewater characteristics (especially nitrogen concentration and pH), CW types, and wetland plants, as presented in Appendix A.

#### 3.2.1. Wastewater Characteristics

The NH_4_^+^ concentration in CWs will impact NH_3_ volatilization, and the NH_3_ flux in CWs is positively related to nitrogen loadings (or NH_4_^+^ concentration). For example, when influent NH_4_^+^ concentration was 136.4, 402.0, and 549.9 mg/L, the estimated NH_3_ volatilization occupied 10.5%, 12.3%, and 20.4 of the nitrogen removal, respectively [101]. Poach et al. [102] demonstrated that a CW receiving partially nitrified wastewater exhibited a low level of NH_3_ volatilization, suggesting that the nitrification process lowered the NH_4_^+^ concentration, decreasing NH_3_ release. Similarly, it has also been shown that the NH_3_ volatilization in marsh–pond–marsh systems could be reduced by decreasing the NH_4_^+^ concentration in wastewater [12]. 

The pH value of wastewater can also impact NH_3_ volatilization. NH_3_ volatilization is a physicochemical process in which NH_4_^+^ is in equilibrium between the gaseous and hydroxyl forms [102]:(2)$$NH_{3\;}\left(aq\right)+H_{2}O=NH_{4}^{+}+OH^{-}$$

In this reaction, alkaline pH favors the presence of the aqueous form of NH_3_ in the solution, while the ion form dominates in acidic or neutral pH conditions. The volatilization losses of NH_3_ in flooded soils and sediments are insignificant when the pH value is below 7.5, and the losses are generally not serious when the pH value is below 8.0 [103]. If the pH value is above 8.0, more nitrogen tends to be lost in the form of NH_3_ in the aquatic system. For example, high amounts of NH_3_ release were observed in pilot hybrid CWs under high pH values; specifically, in CWs with a pH value ranging from 8.4–12.3, NH_3_ volatilization accounted for 5.9–23.1% of TN removal, while in the same CWs with a pH value of 6.9, the proportion was only 1.7% [99].

#### 3.2.2. CW Types

The FWS CWs and open ponds exhibited higher NH_3_ volatilization than SSF CWs and marshes. NH_3_ volatilization occupied 9–44% of the TAN removal in FWS CWs, but the proportion was only 1–18% in SSF CWs [11]. VanderZaag et al. found that SSF CWs maintained healthier canopies and had more aboveground phytomass and significantly higher canopy densities than FWS CWs. The smaller boundary layer above the surface of FWS CWs than in SSF CWs increases the turbulent mixing and NH_3_ volatilization [11]. In a marsh–pond–marsh CW, pond sections produced rates of NH_3_ volatilization greater than 36 mg/m^2^/h, while marsh sections produced rates of less than 16 mg/m^2^/h at nitrogen loads greater than 15 kg/ha/d [104]. It was inferred that the algae photosynthesis created high pH values (by consuming CO_2_) during the day time in pond sections, which promoted the NH_3_ volatilization [12,24].

#### 3.2.3. Plant Species and Richness

Most plants are able to absorb various forms of nitrogen, especially those that have adapted to the presence of nitrogen. Different plant species exhibited different forms of nitrogen uptake preferences, depending on the form of nitrogen available in the soil [24]. Different plant species may absorb different amounts of nitrogen leading to various levels of NH_3_ volatilization. High plant species richness tends to reduce NH_3_ volatilization and the plant composition showed greater effects than species richness. For example, Luo et al. [63] indicated that high plant diversity can reduce NH_3_ volatilization in floating CWs. The communities with *R. japonicus* showed lower NH_3_ volatilization (8.80 μg/m^2^/h) than communities without this species (23.6 μg/m^2^/h) [105]. As NH_3_ volatilization is positively related to NH_4_^+^ content in wastewater, it is probable that some plant species have a well-developed root system, absorbing more NH_4_^+^ in the system, thus reducing NH_3_ volatilization [105]. Communities with high species richness may absorb more NH_4_^+^ and utilize more NO_3_^-^ and produce higher biomass due to niche differentiation (competition) among species. Plants with a higher biomass absorb more NH_4_^+^ and release more H^+^ to maintain the ion balance (between the plant and soil environment), therefore lowering the pH of the water [63]. This explains why the high species richness can mitigate NH_3_ volatilization.

In general, NH_3_ volatilization from CWs is influenced by wastewater characteristics, CW types, wetland plants, etc. NH_3_ volatilization can be reduced by adjusting the wastewater characteristics, such as lowering the nitrogen concentrations and pH value and selecting SSF CWs rather than FWS CWs. Moreover, plant species are also important. In addition to the factors mentioned above, environmental conditions such as air temperature, humidity, and wind speed are likely to impact NH_3_ volatilization [100]. As an important gas product of nitrogen recycling, NH_3_ volatilization is as important as N_2_O emission in CWs, as it accounts for higher proportions of nitrogen removal. It is necessary to simultaneously explore the GHG and NH_3_ emission from CWs, avoiding the increase in one pollutant as a result of reducing another pollutant.

### 3.3. VOC Emission from CWs

VOCs are a diverse group of substances that can react with O_3_, hydroxyl radicals (·OH), and NO_3_^−^, which consequently influence atmospheric chemistry. Globally, VOC emission is estimated to be dominated by biogenic sources [106]. To date, studies on VOCs from CWs have mainly focused on substances, including isoprene (C_5_H_8_), monochlorobenzene (MCB), benzene, and methyl tert-butyl ether (MTBE). C_5_H_8_ can participate in aerosol formation and contribute to the greenhouse effect. The global emission of C_5_H_8_ was estimated to be around 500 Tg C/yr, roughly equivalent to the flux of CH_4_ into the atmosphere, making it the most important hydrocarbon emitted by plants [107]. Most aquatic plants emit high amounts of C_5_H_8_ because the isoprenoid biosynthesis requires large amounts of phosphorylated intermediates [108]. The average emission potential of C_5_H_8_ from a boreal fen was reported to be 680 μg/m^2^/h [106]. However, research showed that reconstructed wetlands (a measure to treat urban and industrial wastes in polluted areas) planted with *P. australis* for treating municipal wastewater would not contribute to high hydrocarbon loads, especially C_5_H_8_ [108]. For MCB volatilization, researchers have different opinions on the role of MCB in CWs. Keefe et al. [109] attributed the primary part of VOC (MCB, 1.4-DCB etc.) removal from a CW to volatilization based on model simulations. Meanwhile, MCB volatilization accounted for only 2–4% of the overall removal and volatilization is a subordinate elimination process for MCB in HSSF CWs [110]. The MCB degradation by bacteria may be the dominant removal process rather than volatilization [111]. As for the treating of benzene- and MTBE-contaminated wastewater, different CWs exhibited different volatilization capacities. Benzene and MTBE via volatilization accounted for 5.6% and 2.4% of the overall contaminant mass loss in common reed CWs, respectively [112]. Volatilization fluxes of benzene and MTBE were low (<5% of total removal) in the CWs [113]. All these studies agree that microbial degradation and plant uptake play dominant roles in the removal process of benzene and MTBE. However, higher MTBE volatilization was observed in the root mat system (i.e., CWs only supported by the densely woven root bed) (24%) compared to conventional CWs (<5% total removal) [113], suggesting that the direct contact of aqueous and gaseous phases may facilitate the MTBE volatilization in root mat systems. The volatilization flux of benzene and MTBE varied with the seasons and the highest fluxes were observed in summer [114]. 

In general, the fate of VOCs in CWs involved multiple processes, including rhizosphere microbial degradation, surface volatilization, plant uptake, etc. These processes can be impacted by the physical and chemical properties of the contaminant (e.g., vapor pressure, air–water partition coefficient, and octanol–water partition coefficient), and environmental conditions, resulting in different levels of VOC emission. VOCs are a group of pollutants that include various types of substances that can participate in complex atmospheric chemical reactions. Few studies have reported the volatilization of VOCs from CWs and we suggest that more exploration should be conducted when using CWs to treat wastewater containing VOCs in the future.

### 3.4. H_2_S Emission from CWs

In wetlands, sulfur is transformed by microbiological processes. Sulfate reduction is accomplished by anaerobes such as *Desulfovibrio* spp. [16]. Adelere and Uduoghene [114] found that, in contrast to CH_4_ and CO_2_ emission (38~54%), the CO and H_2_S emission only accounted for ~0.03% and ~0.14% of the total biogas production in CWs. They inferred that CO and H_2_S can be converted to H_2_CO_3_ and H_2_SO_4_ at the stage of the biological remediation process. The sulfur in influent wastewater is usually in the form of sulfate in an oxidizing environment, but in the form of sulfide in a reducing environment. Sulfates are highly soluble under all temperature and pH conditions, while the solubility of sulfides depends on pH. In an acidic pH (<6), sulfide will be present as H_2_S [115]. The H_2_S emission from CWs also exhibits seasonal variations. For instance, the H_2_S emission from CWs during the rainy season (i.e., September) was as high as 2545 µg/L and only 900 µg/L in the dry season (i.e., March) [116]. Machemer et al. [117] also indicated that the H_2_S emission from the substrate surface of CWs was higher in summer (150 nmol/cm^2^/d) than in winter (0.17–0.35 nmol/cm^2^/d) when CWs were used to remove heavy metals from acid mine drainage. The concentration of SO_4_^2−^ in water varied with microbial reduction, which is a process highly impacted by seasonal temperature. High temperatures in summer and fall accelerate the activity of microcosms to decompose organic matter, resulting in high levels of H_2_S [116]. The lower level of H_2_S observed in winter was because the SO_4_^2−^-reducing bacterial activity was inhibited by the low temperature and H_2_S was trapped in the substrate by the freezing surface [117]. Though H_2_S emission only occurs in CWs for treating specific wastewater (e.g., acidic drainage), it is an important part of the biogeochemical cycle of sulfur and should be a concern.

## 4. Conclusions

The multiple gaseous emissions from CWs are commonly impacted by the structure and component of CWs, influent water characteristics, and environmental parameters. The present knowledge can be concluded as follows: (1) To control CH_4_ and N_2_O emissions, HSSF CWs are recommended rather than FWS and VSSF CWs. Biochar is a promising substrate amendment to mitigate N_2_O emission from CWs. The use of specific plant species is recommended rather than increasing the plant species richness. An influent C/N ratio of 5 will probably result in lower N_2_O emissions. High salinity reduces the removal efficiency of pollutants and GHG (mainly CH_4_ and CO_2_) emissions. Additionally, high environmental temperatures promote the microbial activity and development of plants leading to high emissions of CH_4_ and N_2_O; conversely, relatively lower emissions occur in low temperatures. (2) The NH_3_ volatilization from CWs positively relates to the influent nitrogen concentration and pH. The SSF CWs are preferred to reduce NH_3_ volatilization rather than FWS CWs. High plant species richness will probably reduce NH_3_ volatilization, and specific plant species play an important part in alleviating NH_3_ volatilization. (3) VOCs volatilization from CWs is impacted by the physiochemical properties of contaminants and environmental conditions. (4) The H_2_S emission from CWs is mainly affected by the pH value of the system and seasonal variations.

This study provides conclusive suggestions for the design and operation of CWs from the perspective of simultaneously removing pollutants and abating multiple gaseous emissions in CWs. Meanwhile, we have made specific recommendations for future explorations and investigations as follows: (1) The conclusions illustrated in this study are based on previous studies, which were mainly conducted in small- and medium-scale experiments. We suggest that long-term field investigations during the practical operation of CWs should be strengthened in the future, thereby supplementing conclusions to better support the design of low-emission CWs. Meanwhile, more field investigation is a foundation to provide data support for evaluating the contribution of CWs to global carbon emissions. (2) It is foreseeable that regulating the parameters discussed in this study can reduce GHG emission to some extent. However, there is still room for improvement with more innovative interventions. For example, simultaneously regulating the anammox and n-damo processes is theoretically prospective to mitigate both CH_4_ and N_2_O emissions from CWs. (3) The majority of available studies focused on GHG emissions or NH_3_ volatilization, respectively. Though the VOC and H_2_S emissions only occur when CWs are applied for treating specific wastewater, the emission mechanisms and regulating strategies need to be further investigated as they do harm the environment and humans. We suggested that the different forms of gases for specific elements should be investigated simultaneously in the future.

## Figures and Tables

**Figure 1 ijerph-20-03876-f001:**
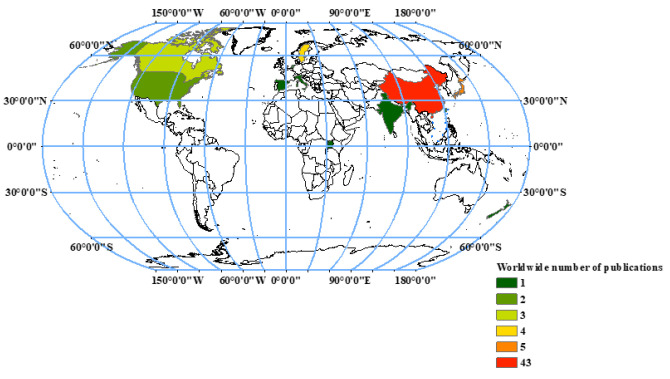
The distribution of publications used in meta-analysis related to GHG emission from CWs.

**Figure 2 ijerph-20-03876-f002:**
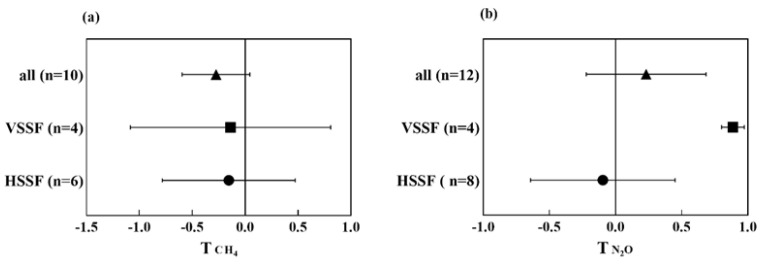
Effect on T index of CH_4_ (**a**) and N_2_O (**b**) emission from different CW types compared to FWS CWs. The n throughout the manuscript refers to samples used in the analysis and the error bars are 95% CIs.

**Figure 3 ijerph-20-03876-f003:**
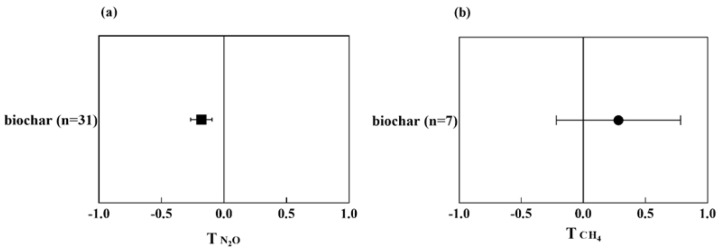
Effect on T index of N_2_O (**a**) and CH_4_ (**b**) emission from biochar-amended CWs compared to gravel-based CWs.

**Figure 4 ijerph-20-03876-f004:**
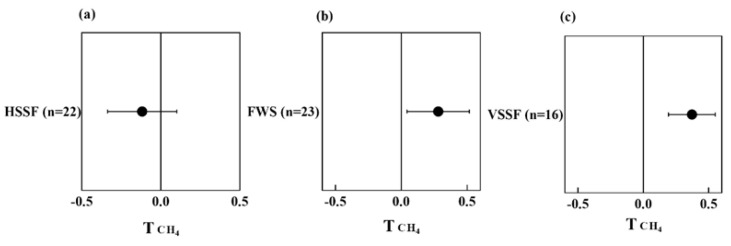
Effect on T index of CH_4_ emission from different types of vegetated CWs, i.e., HSSF (**a**), FWS (**b**), and VSSF (**c**), compared to unvegetated CWs.

**Figure 5 ijerph-20-03876-f005:**
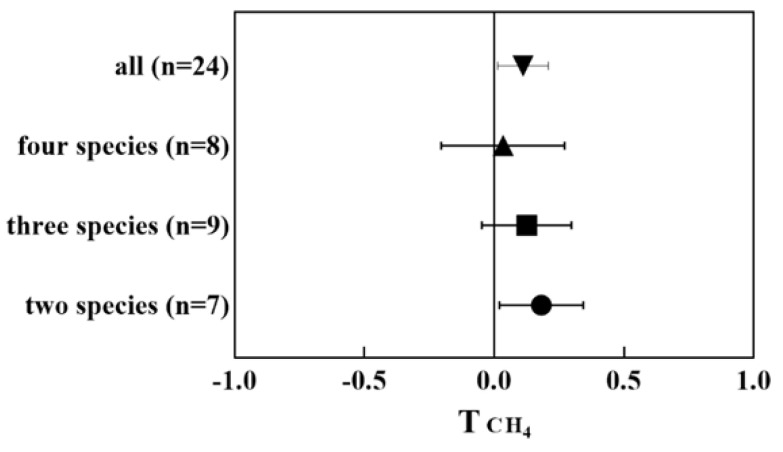
Effect on T index of CH_4_ emission from monoculture CWs compared to polyculture CWs (two, three, and four species).

**Figure 6 ijerph-20-03876-f006:**
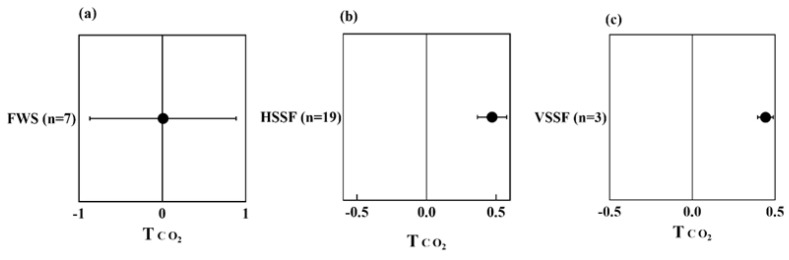
Effect on T index of CO_2_ emission from different types of vegetated CWs, i.e., FWS (**a**), HSSF (**b**), and VSSF (**c**), compared to unvegetated CWs.

**Figure 7 ijerph-20-03876-f007:**
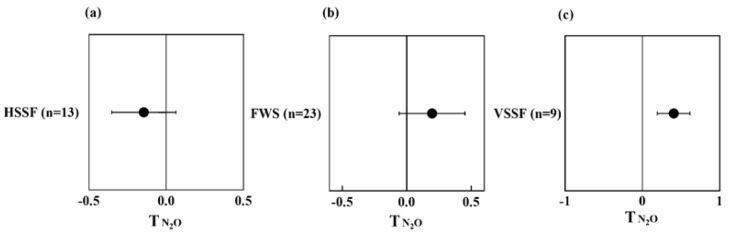
Effect on T index of N_2_O emission from different types of vegetated CWs, i.e., HSSF (**a**), FWS (**b**), and VSSF (**c**), compared to unvegetated CWs.

**Figure 8 ijerph-20-03876-f008:**
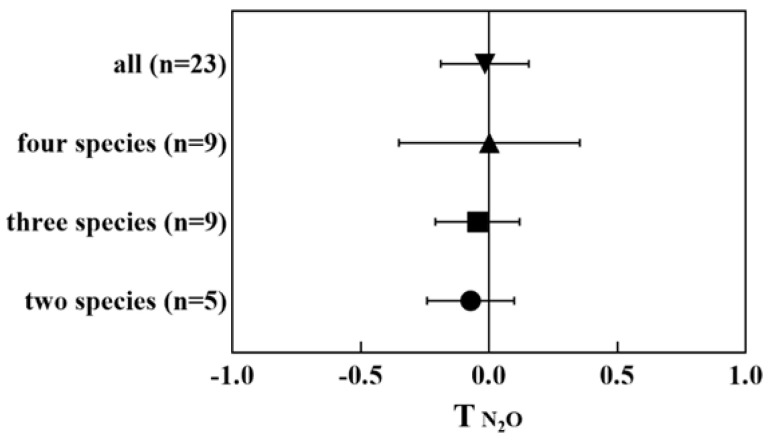
Effect on T index of N_2_O emission from monoculture CWs compared to polyculture CWs (two, three, and four species).

## Data Availability

Data are available on request.

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
