# Peer review of "Factors Influencing Gaseous Emissions in Constructed Wetlands: A Meta-Analysis and Systematic Review"

_ijerph, 2023, doi:10.3390/ijerph20053876_

Round 1
Reviewer 1 Report
This is an interesting manuscript but I found review on this same topic at https://www.researchgate.net/publication/259864324_Greenhouse_Gas_Emission_in_Constructed_Wetlands_for_Wastewater_Treatment_A_Review and the name of the review is “Greenhouse Gas Emission in Constructed Wetlands for Wastewater Treatment: A Review”. The full article is available at https://www.academia.edu/32952017/Greenhouse_gas_emission_in_constructed_wetlands_for_wastewater_treatment_A_review. In the above paper, the authors have reviewed about all the gases that have been reviewed in the current article. The only difference is the other review was done during 2014. Therefore, the current paper is a systematic meta-analysis done in 2022, therefore, is informative.
The article is structured well with understandable language also. Following points need to be clarified before further processing of the article.
1. Write section 2 as methodology and use proper methods for meta-analysis.
2. Certain the conclusion section
3. Draw the limitation of this study
Reviewer 2 Report
This paper conducted a meta-analysis for data collected from current literatures to conclude the impact of the major factors including the structure, plant, substrate types, influent wastewater characteristics and environmental conditions on the greenhouse gas (GHG), NH3, H2S and volatile organic compounds (VOCs) emission from constructed wetlands (CWs).
Detailed comments are summarized below.
1. The paper should be carefully read before submitting for review to avoid general and apparent typos and mistakes.
1) Line 202-203: “pollutantssupportsrt” is a typo.
2) Line 562: what is the “reconstructed wetland”? Is that a typo or a new type of CWs? If that is a new type of CWs, should define it firstly.
2. The language for this paper needs more polish for easier reading. Some words in this paper are inappropriate and some sentences are too redundant.
1) Line 210, the substrates such as biochar, zeolite and other materials shouldn’t be a “concern”, do you want to indicate that these substrates draw more attention?
2) Line 218-224, these sentences are too redundant, should combine them together to indicate that the biochar-amended CWs generate significantly lower level of N2O than gravel-amended CWs.
3) Line 395, what do you mean by “productivity” here? Do you mean the treatment performance?
4) Line 398, “concerned” is not appropriate here.
5) Line 466, should combine these three ranges together to one range.
6) Line 542-543, “concern” is not appropriate here.
3. The major mechanism for GHG, NH3, VOC and H2S generation from CWs should be included in the introduction section.
4. Line 73-74, what kind of GHG does this cited study investigate? And what’s the major conclusion for this study? Should provide more details here.
5. This paper focused on explaining the results collected from the meta-analysis. More discussions should be included to analyze how different factors impact the gas emission from CWs. Also, some conclusions in the discussion part are not rational, the authors shall provide more details to support the conclusions.
1) Line 150-161, the introduction about the structure of different CWs are too brief here. You should provide more details regarding the flow pattern, water saturation conditions and other factors for different CWs. For example, you can not just say the FWS CWs and HSSF CWs differ in water level. You need to demonstrate what the water level is in these two kinds of CWs.
2) Line 162-169, should provide more information about how the CWs structure impact the oxidation-reduction potentials (ORPs) and how the different ORPs impact the microbial community and the greenhouse gas emission.
3) Line 171-171, how does ORP relate to the anaerobic condition? Because of the denitrification process?
4) Line 179-180, why does the subsurface CWs provide more anoxic condition? Because the saturated condition inhibits the diffusion of air? Should provide more details here.
5) Line 188-192, you cannot draw the conclusion that “SSF CWs tend to perform better pollutant removal efficiency than FWS CWs” by indicating that VSSF CWs showed higher NH4+ removal and lower NO3- removal performance than FWS and HSSF CWs. You need to compare the TN removal performance in these two kinds of CWs and provide data concluded from other literatures.
6) Line 226-228, adsorption cannot decrease the substrate availability, it can only concentrate the NH4+/NO3- at the surface of biochar and can not remove them. The biofilm should also be able to remove these attached chemicals via nitrification/denitrification and generate the greenhouse gas.
7) Line 232-237, what are the major functions of these denitrifying bacteria? The reduction of NO3- to N2O or the conversion of N2O to N2?
8) Line 238-242, why do you mention the impact of biochar addition on the GHG emission from aerated CWs in an individual paragraph? It looks incongruent with the previous paragraph. If you want to demonstrate that biochar amendment is not able to reduce the N2O emission potential in all types of CWs, you should use one sentence to connect this paragraph with the previous content at the beginning of this paragraph. And you should provide explanations for the insignificant impact of biochar amendment on the GHG emission from aerated CWs.
9) Lines 243-245, how does biochar promote the interspecies electron transfer between methanogens and anaerobic bacteria?
10) Line 256, How does Mn ore inhibit the CH4+ production and intensify the CH4+ oxidation?
11) Line 302-308, Do both these two plant species have high O2 production and diffusion ability? And are the inhibition of methanogens and promotion of the methanotroph with the addition of these two plants in CWs majorly attributed to their high O2 production ability?
12) Line 314-318, why does poly-culture system contain lower abundance of methanotrophs and higher amount of methanogens? Should provide explanation here.
13) Line 355-358, although carbon released from plant can promote the denitrification process and reduce the N2O emission, the oxygen generated from plant can also inhibit the denitrification process and promote the N2O emission. You should consider more conditions here to explain the reduction of N2O generation with vegetation in CWs. For example, when the influent C/N ratio is not sufficient to support the full denitrification, the carbon source may be a more important factor than oxygen level to control the denitrification process. So the release of carbon from plant may reduce the N2O production. However, when the influent C/N ratio is high, oxygen level may be the dominant factor controlling the denitrification and different result may be observed.
14) Lines 380-385, you just demonstrated that different denitrifying bacteria was observed in CWs with different plant, however, how does that impact the N2O emission? Should provide more details.
15) Line 403-404, why can’t the increasing influent wastewater C/N ratio enhance the TN removal in CWs?
16) Line 414-415, should provide more details about how carbon loading impact the CO2 production from CWs.
17) Lines 435-436, the increased temperature should promote the growth of both methanogens (CH4 generation) and methanotroph (CH4 removal). Does the temperature have different contribution for these two kinds of microorganism?
18) Lines 440-443, you can not only discuss the overall denitrifying bacteria abundance here. The production of N2O from CWs should have higher relationship with the (nirS+nirK)/nosZ or norB/nosZ, you should discuss more about the impact of temperature on denitrifying microorganism containing these functional genes.
19) Lines 504-505, should provide explanations for “SSF CWs tend to maintain healthier plant canopies than FWS CWs”.
20) Lines 524-526, why does only the utilization of NO3- in CWs with high species richness can produce more biomass? What do you mean by different niches? And why does the adsorption of NH4+ can release H+? Should provide more explanations here.
21) Line 562, what do you mean by “conventional CWs”? Is that the CWs filled with traditional materials? What’s is the difference between conventional CWs and novel CWs? Should define it here.
